# Shannon Entropy: An Econophysical Approach to Cryptocurrency Portfolios

**DOI:** 10.3390/e24111583

**Published:** 2022-11-01

**Authors:** Noé Rodriguez-Rodriguez, Octavio Miramontes

**Affiliations:** Departamento de Sistemas Complejos, Instituto de Física, Universidad Nacional Autónoma de México, Ciudad de México C.P. 04510, Mexico

**Keywords:** cryptocurrencies, econophysics, entropy, portfolio uncertainty, heavy-tailed distributions

## Abstract

Cryptocurrency markets have attracted many interest for global investors because of their novelty, wide on-line availability, increasing capitalization, and potential profits. In the econophysics tradition, we show that many of the most available cryptocurrencies have return statistics that do not follow Gaussian distributions, instead following heavy-tailed distributions. Entropy measures are applied, showing that portfolio diversification is a reasonable practice for decreasing return uncertainty.

## 1. Introduction

Financial mathematics and financial economics are two scientific disciplines that together are the backbone of modern financial theory. Both disciplines make extensive use of their own models and theories from mathematics and traditional economics, and are aimed mostly towards predictive analysis of markets. On the other hand, physics has historically had an important influence on economic theory; the formalism of thermal equilibrium was inspirational in the development of the theory of economic general equilibrium [1]. In recent years, there has been a renewed interest in the view that economic phenomena share many aspects of physical systems, and they are thus susceptible to being studied under the science of complex systems [2,3,4] and statistical physics [5], giving rise to novel research fields such as econophysics [6].

One of the first successful econophysical approaches was the discovery that stock market fluctuations are not Gaussian, and instead are heavy-tailed. Mandelbrot came to this conclusion when investigating cotton prices [7], followed later by Mantegna when characterizing the Milan stock market [8]. This discovery forced a serious rethinking of the view introduced by Louis Bachelier in the early 20th Century stating that price variations are random and statistically independent [9]; while historically important, this view not entirely appropriate. These prices are the outcome of the concurrent nonlinear action of many economic agents [10], and as such the fluctuations are usually correlated, meaning that the process is not a random Brownian walk at all. In this article, we explore fluctuations in the cryptocurrency market and show that all of the most common coins analyzed fail the Shapiro–Wilk normality test, and as such are best explained by heavy-tailed statistical models; for further reference, see [11,12].

On the other hand, entropy is a fundamental key quantity in physics in both thermodynamics and information theory. It is related to disorder and diversity, and has been previously used in finance, specifically in portfolio selection theory [13]. Acceptance of entropy in economic theory has historically seen a degree of reticence, for instance, that expressed by Paul Samuelson, an influential figure in 20th Century economics [14]. However, has gained wide acceptance lately [15,16], specially with the development of non-equilibrium thermodynamics and complex systems theory [17,18,19,20]. It is well known that portfolio diversification is a good strategy to minimize specific risks, and as such we explore the use of entropy in the cryptocurrency market as a measure of return uncertainty and risk minimization [13,19,21]. It is worth mentioning that recent studies elsewhere have been interested in exploring the use of entropy in the cryptocurrency markets, where heavy-tailed distributions and multi-scaling phenomena have been identified as well [11,12].

When speaking of information theory, its most powerful tools are considered to be information measures (MI); although the most famous MI is Shannon’s entropy, Fisher’s information is worth mentioning. Shannon’s entropy measures missing information based on the Probability Distribution Function (PDF). The counterpart of the Shannon entropy is the Fisher information, which increases as the Shannon entropy decreases. Furthermore, the Fisher information reflects the amount of positive information from the observer, and depends on the PDF as its first derivatives. For this reason, the Fisher entropy is a local measure, while the Shannon entropy is a global measure. There have been studies elsewhere [22] showing that from the market point of view, minimizing risk is equivalent to minimizing Fisher’s information; the present study provides empirical results that contribute to these perspectives.

## 2. Methods, Data and Analysis

In this article, we use the historical daily prices of 18 cryptocurrencies as presented in CryptoDataDowload [23] spanning at least 3 years. These are: Basic Attention Token (BAT), Bitcoin Cash (BCH), Bitcoin (BTC), Dai (DAI), Eidoo (EDO), Eos (EOS), Ethereum Classic (ETC), Ethereum (ETH), Metaverse ETP (ETP), Litecoin (LTC), Neo (NEO), OMG Network (OMG), Tron (TRX), Stellar (XLM), Monero (XMR), Verge (XVG), Ripple (XRP), and Zcash (ZEC) [24,25].

We used the opening and closing prices of the cryptocurrencies, quoted in US Dollars from 16 October 2018 to 31 December 2021, for a total of 1172 observations per cryptocurrency, to calculate daily returns. First, the distributions of the daily returns of each cryptocurrency were statistically characterized through normality tests and parametric adjustments of heavy-tailed distributions. Subsequently, we performed an analysis in which entropy functions were derived, similarly as in Dionisio et al. (2006), [13], Ormos and Zibriezky (2014) [19], and Mahmoud and Naoui (2017) [26]. From the set of 18 cryptocurrencies, the assets were randomly selected to compose investment portfolios, where the only premise used was that the proportion invested in each asset was $\frac{1}{N}$, with *N* being the number of assets in the portfolio. In order to consistently compare entropy to standard deviation, normal entropy was used, as it is a function of variance. It is important to mention that the time period chosen in this study was arbitrary; while it includes fluctuations due to the COVID-19 pandemic (see [11,27,28]), it certainly does not include recent events that have influenced the behavior of the cryptocurrency markets, such as the war in Eastern Europe and the increased economic rivalry between the USA and China.

In the following subsections, we review several useful concepts related to entropy functions: the discrete entropy function, the continuous entropy function, entropy as a measure of uncertainty, comparison between entropy and variance, and investment portfolios.

The appendices contain the methods used for the analysis of this article. Because the Shannon entropy is only a function of the PDF for the continuous case, Section A.1 shows how the problem should be treated when the PDF is not known and only finite experimental data are available. Section A.1.1 provides a complete solution for entropy estimation by means of the equidistant histogram method. Section A.1.2 shows how the PDF is found via the Kernel method. Section A.1.3 explains how the PDF is found via the parametric method. Furthermore, Section A.1.4 explains the max log-likelihood method that is mentioned throughout the article. Finally, Section A.2 shows the functional form of the distributions used for the parametric fits.

### 2.1. Discrete Entropy Function

Let *X* be a discrete random variable, $\left\{A_{1},A_{2},A_{3},\ldots A_{n}\right\}$ be a set of possible events, and the corresponding probabilities $p_{X}\left(x_{i}\right)=Pr\left(X=A_{i}\right)$, with $p_{X}\left(x_{i}\right)\geq 0$ and $\sum_{i=1}^{n}p_{X}\left(x_{i}\right)\;=\;1$. The generalized discrete entropy function, or Rényi entropy [19,29], for the variable *X* is defined as
(1)$$H_{\alpha }\left(X\right)=\frac{1}{1-\alpha }log\left(\sum_{i=1}^{n}p_{X}{\left(x_{i}\right)}^{\alpha }\right)$$
where α is the order of the entropy and α≥0. This order can be considered as a bias parameter, where α≤1 favors rare events and α≥1 favors common events [30]. The base of the logarithm is 2.

When α=1, this indicates a special case of generalized entropy that assumes ergodicity and independence, which the generalized case does not. However, substituting into (Equation 1) results in division by zero. By means of L’Hôpital’s rule, it can be shown that when α tends to 1, we have the Shannon entropy:(2)$$H_{1}\left(X\right)=-\sum_{i=1}^{n}p_{X}\left(x_{i}\right)log\left(p_{X}\left(x_{i}\right)\right)$$

The Shannon entropy produces exponential equilibrium distributions, while generalized entropy produces power law distributions.

### 2.2. Continuous Entropy Function

Let *X* be a continuous random variable that takes values of R and let $p_{X}\left(x\right)$ be the density function of the random variable. Continuous entropy is defined as
(3)$$H_{\alpha }\left(X\right)=\frac{1}{1-\alpha }ln\int p_{X}{\left(x\right)}^{\alpha }$$

Note that the logarithm bases of (Equation 1) and (Equation 3) are different. Although the entropy depends on the base, it can be shown that the value of the entropy changes only by a constant coefficient for different bases.

When α=1, we have the Shannon entropy for the continuous case
(4)$$H_{1}\left(X\right)=-\int_{}^{}p_{X}\left(x\right)ln\left(p_{X}\left(x\right)\right)dx$$

The properties of discrete and differential entropy are similar. The differences are that the discrete entropy is invariant under variable changes and the continuous entropy is not necessarily; furthermore, the continuous entropy can take negative values.

### 2.3. Entropy as a Measure of Uncertainty

According to Shannon (1948) [31], an uncertainty measure $H(p_{X}\left(x_{1}\right),p_{X}\left(x_{2}\right),\ldots ,p_{X}\left(x_{n}\right))$ must satisfy the following:*H* must be continuous on $p_{X}\left(x_{i}\right)$, with i=1,…,n.If $p_{X}\left(x_{i}\right)=\frac{1}{n}$, *H* must be monotone increasing as a function of *n*.If an option is split into two successive options, the original *H* must be the weighted sum of the individual values of *H*.

Shannon showed that a measure that satisfies all these properties is (Equation 2) multiplied by any positive constant (the constant sets the unit of measure). Among the properties that make it a good uncertainty choice are:H(X)=0 if and only if all but one of $p_{X}\left(x_{i}\right)$ are zero.When $p_{X}\left(x_{i}\right)=\frac{1}{n}$, i.e., when the discrete probability distribution is constant, H(X) is maximum and equal to log(n).H(X,Y)≤H(X)+H(Y), where the equality holds if and only if *X* and *Y* are statistically independent, i.e., $p\left(x_{i},y_{j}\right)=p\left(x_{i}\right)p\left(y_{j}\right)$.Any change towards the equalization of the probabilities $p_{X}\left(x_{i}\right)$ increases *H*.H(X,Y)=H(X)+H(Y|X); thus, the uncertainty of the joint event (Y|X) is the uncertainty of *X* plus the uncertainty of *Y* when *X* is known.H(Y)≥H(Y|X), which implies that the uncertainty of *Y* is never increased by knowledge of *X*, and decreases unless *X* and *Y* are independent, in which case it does not change.

### 2.4. Comparison between Entropy and Variance

Ebrahimi et al. (1999) [32] showed that entropy can be related to higher order moments of a distribution, and thus can offer a better characterization of $p_{X}\left(x\right)$, as it uses more information about the probability distribution than the variance (which is only related to the second moment of a probability distribution).

The entropy measures the disparity of the density $p_{X}\left(x\right)$ of the uniform distribution. That is, it measures uncertainty in the sense of using $p_{X}\left(x\right)$ instead of the uniform distribution [26], while the variance measures an average of distances from the mean of the probability distribution. According to Ebrahimi et al. [32], both measures reflect concentration, although they use different metrics. The variance measures the concentration around the mean and the entropy measures the density diffusion regardless of the location of the concentration. Statistically speaking, entropy is not a mean-centered measure; it takes into account the entire empirical distribution without concentrating on a specific moment. In this way, the entire distribution of returns can be considered without focusing on one in particular [33]. The discrete entropy is positive and invariant under transformations, while the variance is not. In the continuous case, neither the entropy nor the variance are invariant under one-to-one transformations [13,26]. According to Pele et al. (2017) [34], entropy is strongly related to the tails of the distribution; this feature is important for distributions with heavy-tails or with an infinite second-order moment, where the variance is obsolete. Furthermore, the entropy can be estimated for any distribution without prior knowledge of its functional form. The authors mentioned above found that heavy-tailed distributions generate low entropy levels, while light-tailed distributions generate high entropy values.

### 2.5. Investment Portfolios

A portfolio or investment portfolio is simply a collection of assets. They are characterized by the value invested in each asset. Let $w_{i}$ be the fraction invested in asset *i* with i=1,2,…,n; the required constraint is that
(5)$$\sum_{i=1}^{n}w_{i}=1$$

We define the return $R_{i}$ of a common share *i* during a certain period as
(6)$$R_{i}=\frac{P_{1,i}-P_{0,i}}{P_{0,i}}$$
where $P_{0,i}$ is the price of stock *i* at the beginning of the period and $P_{1,i}$ is the price of *i* at the end of that period. This return is the historical return or ex post return, whereas the total portfolio return is simply the weighted average of the expected returns of the individual securities in the portfolio:(7)$$R_{T}=\sum_{i=1}^{n}w_{i}R_{i}$$

The risk of investment portfolios can be divided into specific and systematic risk. Systematic risk is inherent to market uncertainty, and is not diversifiable; usually, the price of an asset is affected by factors such as inflation, economic growth or recession, and fluctuations in the world financial markets. The specific or unsystematic risk corresponds to the risk of an asset or a small group of assets due to its specific characteristics, and is diversifiable. Forming portfolios can reduce specific or unsystematic risk.

Entropy can provide similar information if we define the average mutual information between two random variables *X* and *Y*, where one is the independent variable and the other is the dependent variable, as follows:(8)$$I\left(X,Y\right)=\sum_{i=1}^{n}\sum_{j=1}^{m}p_{X,Y}\left(x_{i},y_{j}\right)log\left(\frac{P_{X,Y}\left(x_{i}|y_{j}\right)}{P_{X}\left(x_{i}\right)P_{Y}\left(y_{j}\right)}\right)$$

for the discrete case and
(9)$$I\left(X,Y\right)=\int_{X}^{}\int_{Y}^{}p_{X,Y}\left(x,y\right)log\left(\frac{P_{X,Y}\left(x|y\right)}{P_{X}\left(x\right)P_{Y}\left(y\right)}\right)$$
for the continuous case. This measure reflects the correlation between *X* and *Y*. Thus H(X)=I(X,Y)+H(X|Y), where I(X,Y) can be compared to systematic risk, while H(X|Y) can be compared to the specific risk.

In practice, standard risk measures such as CAPM beta or standard deviation are calculated on daily or monthly returns. We follow this methodology here as well. Because the returns of the values can take values of a continuous co-domain, we focus mainly on the differential entropy.

The assumption that data and residuals follow a normal distribution is common in both portfolio management and risk analysis. Therefore, in this context, in order to parametrically estimate the entropy of a normal distribution (because information theory measures cannot be directly compared to variance in metric terms), we use
(10)$$NH\left(X\right)=\int p_{X}\left(x\right)log\left(\sqrt{2\pi }\sigma \right)dx+\int p_{X}\left(x\right)\frac{{\left(x-\bar{x}\right)}^{2}}{2\sigma^{2}}dx=log\left(\sqrt{2\pi e}\sigma \right)$$

To calculate the empirical differential entropy, it is only necessary to estimate the Probability Distribution Function (PDF), for which there are essentially three methods: histograms, kernels, and parametrically (see Appendix A for details).

## 3. Results

We present our findings in three parts. First, it is shown that the daily returns of each cryptocurrency do not follow a normal distribution. Second, it is indicated that the daily returns have heavy tails. Finally, the behavior of the empirical and normal entropy is shown as a function of the number of assets in the portfolio. As far as we know, this is the first time that such a large number of cryptocurrencies has been analyzed together; however, see [11,12].

### 3.1. Normal Distribution and Q-Q Plots

Figure 1, made according to Equations (Equation 6) and (Equation 15), shows as an example the histogram of the probability distribution of the daily returns of Bitcoin considering the daily opening and closing price. The blue curve shows the normal distribution with the variance and mean calculated from the empirical data. The red line shows the Kernel density estimator; the cross-validation method was used to determine the bandwidth and the Kernel that maximize the total Log-Likelihood of the data in *X* [35]. Although there are alternative principles, such as MaxEnt or FIM-minimization, for estimating the best PDF, log-likelihood was used for simplicity. Among the kernels used were Gaussian, Epanechnikov, Tophat, exponential, linear, and cosine. Bandwidth was optimized for values between 0 and 1. For all cryptocurrencies, a significant deviation of the distribution calculated with Kernels was found with respect to the normal distribution.

Quantiles are values that divide a probability distribution into equal intervals, with each interval having the same fraction of the total population. Q-Q plots are commonly used to visualize data and comparatively find the type of probability distribution to which a random variable can belong, for example, whether they are Gaussian, uniform, Paretian, exponential distributions, etc. To build a Q-Q plot, the quantiles of the base distribution, or the "theoretical quantiles", are plotted on the *X* axis, while the sample quantiles are plotted on the *Y* axis. If both sets of quantiles come from the same distribution, then the points form a more or less straight line.

Figure 2 shows the normal Q-Q plot of the empirical daily returns of Bitcoin. Across all cryptocurrencies, the normal distribution was found to capture the middles of the data well, but not the tails. In fact, the characteristic behavior of heavy-tailed distributions can be clearly seen by the “S” shape on the graph. This means that compared to a normal distribution there are more data located in the tail of the heavy-tailed distribution.

The Shapiro–Wilks test posits the null hypothesis that a sample comes from a normal distribution. When applying this statistical test to the daily returns of all the cryptocurrencies studied, it was found that the *p*-value is at most on the order of $10^{-18}$. It is established as common practice that if the *p* value is less than 0.05, the hypothesis that the sample is normal should be rejected. In our case, the value of *p* is very small (indeed, almost negligible) for all cryptocurrencies, which strongly indicates that the null hypothesis that daily returns follow a normal distribution [36] should be rejected. Therefore, we accept the alternative hypothesis that the returns do not present normality.

### 3.2. Heavy-Tailed Distributions

Nadarajah et al. (2015) [37] made adjustments to the exchange rate of various fiat currencies (money that is not backed by securities or physical assets such as gold, and is instead backed by the government that issues it) using flexible distributions. They used distributions such as Student’s t, skewed Student’s t, hyperbolic, generalized hyperbolic, generalized lambda, Skew-T, and Gaussian inverse normal to analyse the data. According to the previous section, we must limit ourselves to the study of distributions with long tails. Osterrieder (2016) [24] performed heavy-tailed fits for different cryptocurrencies using Student’s *t*-test, Generalized Student’s t, Hyperbolic, Generalized Hyperbolic, Gaussian Inverse Normal, and Asymmetric Gamma variance distribution. In both studies, they found that all the heavy-tailed distributions provided statistically similar results, although the best fit was found with the generalized hyperbolic distribution. In addition, they concluded that Student’s *t*-test is a good option considering its simplicity. Similarly, Briere et al. (2017) [38] reached similar results when studying parametric adjustments to the returns of the seven most traded cryptocurrencies of 2015. Therefore, we make adjustments here to the daily returns with the Student’s t-distribution and the normal distribution for comparison, although it is light-tailed.

Figure 3 shows the histogram of the probability distribution of Bitcoin’s daily performance along with the adjustments of each distribution, meaning that the log-likelihood is maximized. The log-likelihood value for each fit is shown in Table 1. Note that the log-likelihood is always smaller for the normal distribution, while the heavy-tailed Student’s t-distribution generates good fits.

### 3.3. Portfolio Entropy

A portfolio was generated by adding the 18 cryptocurrencies randomly. We began by adding an asset to the portfolio, then its normal entropy and empirical entropy were calculated using the histogram, Kernel, and parametric methods. Assets continued to be added randomly until the 18 cryptocurrencies studied had all been added. Every time a new asset was added to the portfolio, it was necessary to calculate the normal entropy and the empirical entropy of the total return; i.e., every time an asset was added, the distribution function of the total return was calculated again by means of the three methods presented. Figure 4 shows the adjustments when performing the previous procedure, while Figure 5 shows the normal and empirical entropy against the number of assets in the portfolio.

## 4. Discussion

The yields of the 18 cryptocurrencies were studied, with several of their statistical properties shown in detail. For this, the daily returns were adjusted to different heavy-tailed distributions such as the normal and Student’s t-distributions. Our results show that the distributions of daily yields exhibit statistically significant leptokurtic return distributions with heavy tails, as expected. In particular, Student’s t-distribution adequately describes the data and is a good option considering its simplicity; it can be useful for financial risk management, where it is necessary to calculate the value at risk (VaR) and the expected deficit (ES). The aforementioned results are useful for investment purposes as well. As far as we know, this is one of the few works to investigate the statistical properties of many cryptocurrencies together, beyond just Bitcoin or Ethereum, although see [11,12] as well.

Entropy is used here as a measure of uncertainty in portfolio management. Figure 5 shows that the Shannon entropy behaves similarly, although not exactly the same, as the variance, serving as a measure of risk and to verify the effect of diversification, both of which tend to decrease when including assets in the portfolio. As the number of assets in the portfolio increases, the possible number of states of the system, in this case the portfolio, progressively increases as well, and the uncertainty about this portfolio decreases according to the property H(X,Y)≤H(X)+H(Y). We should note that the normal entropy always takes values greater than the empirical entropy in all cases, which implies that the uncertainty is less than that which would be observed if the returns were normally distributed.

In the study by Ormos and Zibriezky (2014) [19], it was found that the histogram method for calculating the Shannon entropy was more efficient in terms of explanatory and predictive power and exhibited simplicity compared to the Kernel method. We observe that the fits of Figure 4 to the daily total return of the generated portfolios show that the histogram method presents overfitting compared to the empirical data. The small “jumps” observed in the tails with this method are evident in the log–log scaled plot. On the other hand, the best fit is parametric, followed by KDE. In these last two cases, the log-likelihood of the parametric method with Student’s t-distribution is always greater than that of the Kernel Density Estimation (KDE) method, although the log-likelihood of both is always greater than that of the normal distribution. Regardless, diversification is verified with any method. This corroborates and contributes to the empirical findings made by Dionisio et al. (2006) [13], Ormos and Zibriezky (2014) [19], and Mahmoud and Naoui (2017) [26], all of whom observed the diversification of the Shannon entropy, with the difference that these authors verified it in classical assets and did not characterize their distributions. Our results can be useful to investors, traders, and portfolio managers.

## 5. Conclusions

Based in our findings, we conclude that entropy observes the effect of diversification and is a more general measure of uncertainty than the standard deviation. This is the case for the following reasons. (i) It uses more information from the probability distribution, as it uses moments of higher order while the deviation standard only uses the second moment. (ii) It does not depend on any particular distribution, unlike the standard deviation, which eliminates the error introduced by fitting a normal distribution to returns. This becomes evident in non-symmetric distributions with additional non-normal moments. In this way, entropy can capture the complexity of systems without the need for hypotheses that could bias the results. (iii) Entropy is independent of the mean considering any distribution, meaning that it satisfies the first-order conditions. (iv) Entropy meets the requirements that good uncertainty measures must satisfy. (v) Entropy can be used for both metric and non-metric data (quantitative and qualitative data, respectively). For this reason, it can be used as a measure to complement traditional models, thanks to a mean and variance that are more restrictive, when dealing with assumptions that tend not to be verified empirically.

The drawbacks of the potential use of entropy as a measure of uncertainty in portfolios should be mentioned as well, one of them being that it is more complex compared to the more common standard deviation. On the other hand, entropy does not take into account the real values of the variables, meaning that care must be taken when using it in risk analysis and portfolio selection. In addition, there is always statistical bias in data measurements due to the degrees of freedom allowed in an experiment.

Although the present study does not address punctual prediction of the cryptocurrency market, it is illustrative to mention the relative success in this area of the use of nonlinear physics methods to make substantive predictions, among others by the “Prediction Company” founded by two pioneers of chaos theory in 1991 and associated with the Santa Fe Institute. This is the case of Doyne Farmer and Norm Packard [39]. In one of his best-known contributions [40], Farmer mentions the possibility of using “adaptive dynamics” to make predictions in natural time series, which follows the principles of neural networks, more currently known as known as “machine learning”. His study basically consists of the use of systems that learn to classify patterns (output) from a time series (input). The output would ideally be an investment strategy that would aim to maximize profits and the input would be the immediate, ideally real-time, time series of market behavior. Of course, there are recent applications of machine learning and other methods to the prediction of the temporal behavior of cryptocurrencies [41,42].

For the temporal prediction of financial markets, today there are more than 5000 algorithms created by various programmers who publish them in public software libraries [43]. Investors typically use several of these algorithms in combination, although the most popular are usually based on Average True Value (ATR), Relative Strength Index (RSI), Moving Average Convergence/Divergence (MACD), and Exponential Moving Average (EMA) [43]. As can be seen, the use of moving averages is common, as they essentially help to reduce noise or fluctuations in time intervals. Several authors [44,45] have coupled them with regression analysis to form ARMA (autoregressive moving average) models, which in turn are used together with skewed (non-normal) probability distribution functions because, as we have already confirmed here, the PDFs of the financial markets and cryptocurrencies are not normal. It is interesting to note that there are “forecasts” online that suggest the right times, in real time, to sell or buy cryptocurrencies, and that these are mostly based on moving average models [46].

Finally, it is illustrative to mention that cryptocurrency markets have qualitative and quantitative properties similar to traditional financial markets [44,45], which allows us to speak of generic properties between the two, that is, aspects of universality typical of complex systems. Studying the statistical behavior of cryptocurrency markets, a phenomenon we have witnessed from its birthday while having at hand the whole arsenal of contemporary mathematical tools, is a fantastic opportunity to understand the real-world dynamics of a complex economical system, the interactions of the agents involved, and the emergent collective properties that may influence our society. For example, in future cryptocurrency markets, financial markets, e-commerce markets, and energy markets [28,47,48,49,50]). We hope our contribution can be useful in understanding the statistical properties of contemporary market and portfolio modeling.

## Figures and Tables

**Figure 1 entropy-24-01583-f001:**
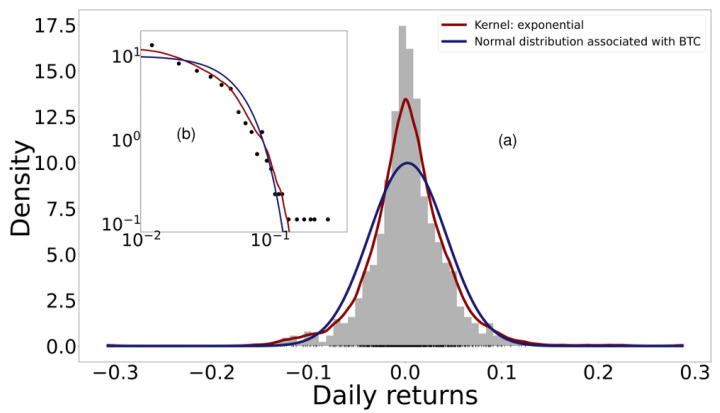
(**a**) Histogram of the probability distribution of the return of Bitcoin. The normal curve is shown in blue, associated with the dataset, with the curve estimated using the Kernel method in red. (**b**) The log–log plot of the same data is depicted in the inset. Notice that the normal distribution tail decays faster in comparison with the kernel, as expected.

**Figure 2 entropy-24-01583-f002:**
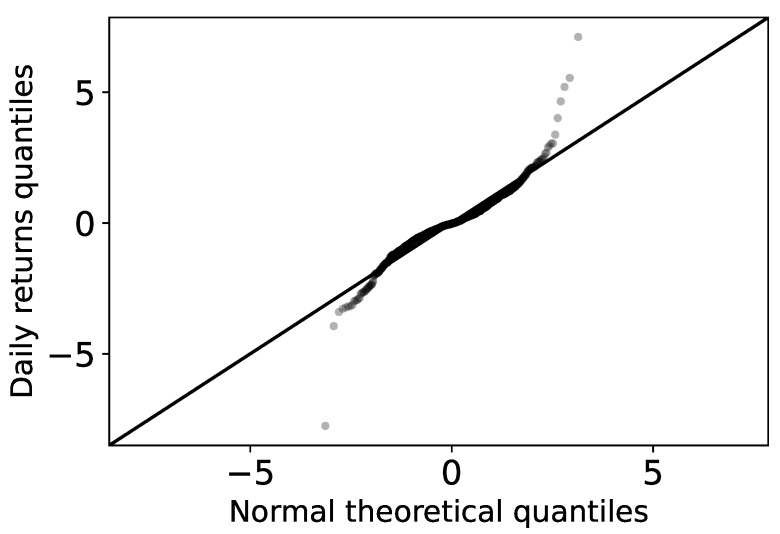
Normal Q-Q chart of daily Bitcoin performance. The theoretical normal quantiles are plotted against the observed quantiles.

**Figure 3 entropy-24-01583-f003:**
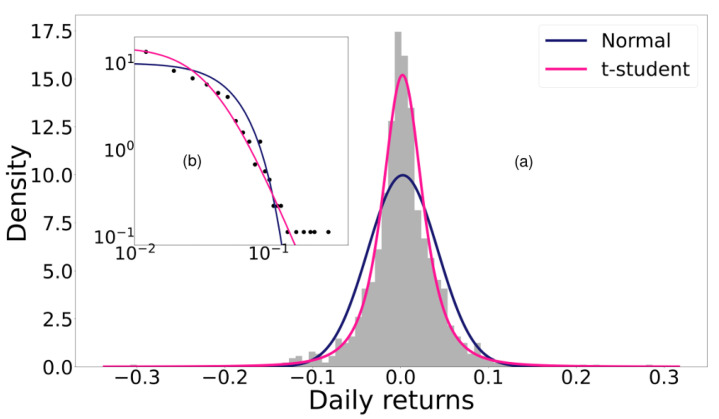
(**a**) Histogram of the probability distribution of the return of Bitcoin with the adjustments of a normal distribution and Student’s t. (**b**) The log–log plot of the same data is depicted in the inset. Notice that the normal distribution tail decays faster in comparison with the Student’s t-distribution, as expected.

**Figure 4 entropy-24-01583-f004:**
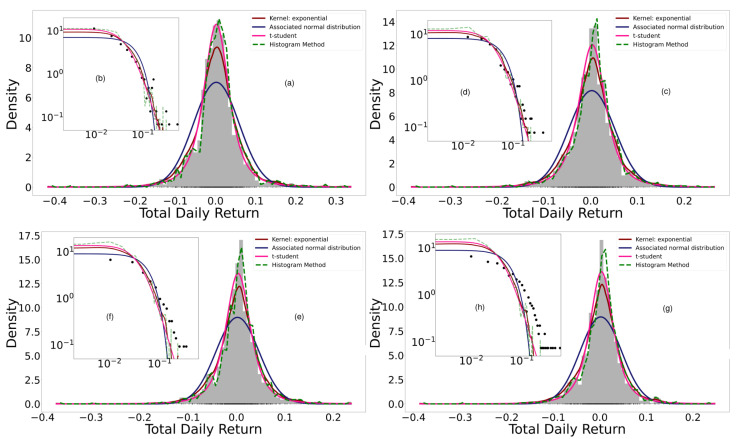
(**a**,**c**,**e**,**g**) correspond to the adjustments of the probability distribution of the total daily returns with n∈{3,8,13,18} assets in the portfolio, respectively, with the kernel, parametric, and histogram methods and the normal curve associated with the data. In (**b**,**d**,**f**,**h**), the same fits are shown in log–log scale.

**Figure 5 entropy-24-01583-f005:**
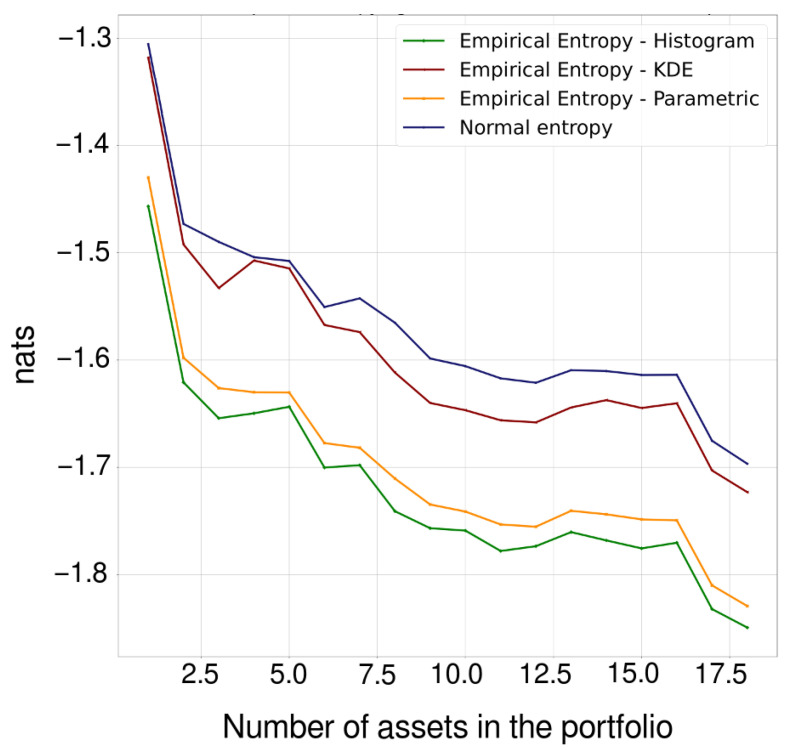
Normal and empirical entropy against the number of assets in the portfolio. The decrease in entropy as the number of assets in the portfolio increases verifies the diversification effect.

**Table 1 entropy-24-01583-t001:** Log-likelihood of the fitted distributions.

Cryptocurrency	Normal	t-Student
BAT	1531	1667
BCH	1555	1815
BTC	2111	2250
DAI	4395	4871
EDO	1239	1518
EOS	1583	1794
ETC	1608	1861
ETH	1831	1915
ETP	1393	1594
LTC	2111	2250
NEO	1631	1740
OMG	1401	1570
TRX	1694	1829
XLM	1590	1808
XMR	1814	1947
XRP	1595	1884
XVG	1305	1396
ZEC	1637	1722

## Data Availability

The data analyzed in this article are freely available from the following source: https://www.cryptodatadownload.com, accessed on 1 January 2022.

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
