# Peer review of "Shannon Entropy: An Econophysical Approach to Cryptocurrency Portfolios"

_entropy, 2022, doi:10.3390/e24111583_

Round 1

Reviewer 1 Report

comments and suggestions in the enclosed file

Author Response

Reviewer 1 comments

This paper by Rodriguez and Miramontes treats the fluctuations of cryptocurrencies over a specific period of time. The main conclusion is that such fluctuations are not normally distributed but they have fat tails. This behavior is expected, and goes along with the first observations in Econophysics about fluctuations in the markets from the 1990s. While this was expected, nevertheless, the article is worth publishing as it gives in great detail additional information by bringing in the idea of entropy, to examine in detail the behavior of the cryptos.

Answer: We very much appreciate this positive comment on our article.

I have the following comments.

The period investigated was 10/2018 to 12/2021. Why was this period chosen? The problem with this period is that for the largest part (10/2018 to 04/2021) there was an almost monotonic increase in the value of BTC with only minor fluctuations. Only the last few months of the period used was a 50in the value of BTC. Wouldn’t it better to choose a period with large fluctuations in the entire domain?

Answer: The period chosen was arbitrary and the ending date is the day we started this investigation, We agree that recent real world facts such as the Ukrainian war, the China-US rivalry, the shadow of recession in Europe and the USA has made markets much more volatile including the cryptocurrency one; but as the reviewer under- stand, there is no way we could have anticipated this turbulence and obviously there is no way to predict how markets will fluctuate over the next winter months. We have included a sentence in the Methods section to explain why this period was chosen.

Line 199-200. The sentence “more data at the extreme of the distribution and less data at the center” is not clear. What is meant by this? Are there more data available at the tails rather than the middle ? If so, is this correct ? I would think that the opposite is true, since there are fewer very large fluctuations and a lot more days with no fluctuations or very small ones. Please explain.

Answer: We agree that this sentence is a bit sloppy and have reworded it. We mean that in the long-tailed distributions there are more data in the tail that in the tail of a comparable normal distribution.

Line 257-259: I think the opposite is true. As more assets are added to a portfolio, there is a larger number of combinations, and therefore a larger number of states in the system. But in such case since by adding more assets the weight of each individual asset goes down, it would be true that the uncertainty would decrease. Please explain here the relationship between the number of states and the uncertainty for the overall portfolio.

Answer: Thank you very much for this correction. Certainly the states of the system increase and the entropy decreases due to property three of section 2.3

Line 290: What is meant by “metric”? Do they mean measurable? Still whatever the meaning of this term, please explain the distinction with a few more words and/or ideas at this point on the Conclusions section.

Answer: Thank you very much for this correction of the use of language. In this case, we take the definition that data can be classified into two categories: metric and non-metric. Metric and non-metric data are also known as quantitative and qualitative data, respectively. Entropy has the ability to work with qualitative data, this is a common method in Machine Learning, where classification uncertainty for categorical data can be treated by means of cross-entropy.

X is defined in Equation 1. X and Y are not defined in Equation 8. Are they also here two random variables? Maybe their definition can be repeated here.

Answer: True, the definition of Y is not given. Thanks for the observation. In this case X and Y are two random variables where one is the independent variable and the other is the dependent variable. This measure reflects the correlation between X and Y.

Figure 1: The x-axis is daily returns. It looks like ∆x is not 1 day in this plot. What is ∆x here? Is there some boxing done to plot the data here? Can this be mentioned quantitatively?

Answer: For graph 1, the x-axis was obtained through equation (6), where P0,i was taken as the opening price and P1,i as the closing price per day. Subsequently, a normalized histogram was made with equation (A4) and (A5).

Figure 4: practically everything is this figure is not readable, Characters are too small, on the x an y axes, and also in the inserts. They should be made readable with larger size fonts.

Answer: We have modified the figures to make them more readable by enlarging the fonts.

By the other hand” should be “On the other hand” everywhere in the text.

Answer: Corrected

The Figure 1”, “The Figure 2”, should be “Figure 1”, “Figure 2”, just like

correctly it is reported “Figure 3”

Answer: Corrected

data analyzed in this article is freely” should be “data analyzed in this article

are freely”

Answer: Corrected

Summarizing, the paper is worth publishing after the authors response to the above comments and make the necessary corrections so that the text is readable everywhere.

Answer: We very much appreciate this comment

Reviewer 2 Report

I encourage the authors to clearly define the scientific result they present in their work.

The fact that the probability distributions of the yields of securities (not just cryptocurrencies) do not follow the Gauss distribution is a well-known observation. Likewise, the relationship between Information and Entropy and the measurement of uncertainty in the markets. Fisher's information, for example, is significantly related to risk in quantum market models (Minimum Fisher information principle).

I understand that the work submitted for review uses certain known methods. In that case, it would be appropriate to include a robust literature review, taking into account the latest results in market modeling.

Calculations alone are not enough. It is not very clear what they are made for.

I have mixed feelings about this paper.

To summarize:
1) Authors should prepare a solid literature review in terms of the methods they use.
2) Justify why their work is important to understanding markets (or complex systems in general). As it stands, it looks like a set of standard statistical calculations.

Author Response

Reviewer 2 comments

I encourage the authors to clearly define the scientific result they present in their work.

Answer: Thank you. We have modified the discussion section on the light of this comment. We hope it is more clear now.

The fact that the probability distributions of the yields of securities (not just cryptocurrencies) do not follow the Gauss distribution is a well-known observation. Likewise, the relationship between Information and Entropy and the measurement of uncertainty in the markets. Fisher’s information, for example, is significantly related to risk in quantum market models (Minimum Fisher information principle).

Answer: Indeed. We mentioned already in the in the introduction that stock markets analyzed elsewhere show non-Gaussian distributions. However it is the first time, to the best of our knowledge, that a number of cryptocurrencies are together analyzed to show this. We now include references to Fisher’s work.

I understand that the work submitted for review uses certain known methods. In that case, it would be appropriate to include a robust literature review, taking into account the latest results in market modeling.

Answer: We have strengthened the literature review to make it more complete and robust. More references about the methods have been added.

Calculations alone are not enough. It is not very clear what they are made for.

Answer: We appreciate this comment since it is helpful to improve the text. We added more text in the methods to explain why the calculations were done.

I have mixed feelings about this paper. To summarize: 1) Authors should prepare a solid literature review in terms of the methods they use. 2) Justify why their work is important to understanding markets (or complex systems in general). As it stands, it looks like a set of standard statistical calculations.

Answer: We strengthened the literature review. We now include more text in the introduction and the conclusions sections about why this work is important for understanding markets and the behavior of complex systems in general

Round 2

Reviewer 2 Report

The authors improved the works.  In my opinion, it is ready for publication